# Pancuronium Bromide for Chemical Immobilization of Adult Nile Crocodiles (*Crocodylus niloticus*): A Field Study

**DOI:** 10.3390/ani13101578

**Published:** 2023-05-09

**Authors:** Lionel Schilliger, Chawki Najjar, Clément Paillusseau, Camille François, Frédéric Gandar, Hela Boughdiri, Marc Gansuana

**Affiliations:** 1Argos SpéNac Veterinary Clinic, 100 Bvd de la Tour Maubourg, 75007 Paris, France; paillusseauclement@gmail.com (C.P.); mille.francois21240@gmail.com (C.F.); 2Veterinary Clinic of El Hidhab, 1st Salaheddine Ayoubi, Hidhab 2082, Tunisia; chawkinajjar.dv@gmail.com; 3Alvetia Veterinary Clinic, 149 Route de Guentrange, 57100 Thionville, France; fredericg.alvetia@gmail.com; 4Djerba Explore Park, Djerba Midoun 4116, Tunisia; hela.boughdiri@gmail.com (H.B.); marc.gansuana@gmail.com (M.G.)

**Keywords:** crocodilians, chemical immobilization, Nile crocodile, pancuronium bromide, neostigmine, non-depolarizing neuromuscular blocking agents

## Abstract

**Simple Summary:**

Due to their significant size and aggressiveness, the capture of adult crocodiles carries significant risk, both in terms of stress and injuries to themselves and to operators. Neuromuscular blocking agents act by inducing flaccid muscle paralysis, thereby reducing the physical and chemical risks associated with transportation and anesthesia. Pancuronium bromide and its antagonist, neostigmine methylsulfate, have been successfully used in juvenile and subadult saltwater crocodiles (*Crocodylus porosus*), but their applications in larger animals (body weight > 230 kg or total length > 3.8 m) or in Nile crocodiles (*Crocodylus niloticus*) have never been described. We trialed a dose recommendation in nine Nile crocodiles using pancuronium bromide that was originally established for small- and medium-sized saltwater crocodiles, the effect of which can be reversed using neostigmine. We found that the recommended dose caused a prolonged recovery time in adult male Nile crocodiles, and we propose a weight-independent dose for Nile crocodiles with body weight ≥ 300 kg or total length ≥ 4.0 m, which we successfully trialed in 32 animals.

**Abstract:**

(1) Background: Pancuronium bromide is a neuromuscular blocker used for immobilizing crocodiles that can be reversed with neostigmine. A recommended drug dose has only been established for saltwater crocodiles (*Crocodylus porosus*), mostly based on trials in juveniles and subadults. After trialing a dose recommendation in a small cohort of nine Nile crocodiles (*Crocodylus niloticus*), we developed and applied a new dose recommendation for large adult Nile crocodiles. (2) Methods: we trialed and adapted a pancuronium bromide (Pavulon 4 mg/2 mL) dose in Nile crocodiles originally established for saltwater crocodiles and applied the new dose for the immobilization of 32 Nile crocodiles destined for transport. Reversal was achieved with neostigmine (Stigmine 0.5 mg/mL). (3) Results: Nine crocodiles were included in the trial phase; the induction time was highly variable (average: 70 min; range: 20–143 min), and the recovery time was prolonged (average: 22 h; range: 50 min–5 days), especially in large animals after reversal with neostigmine. Based on these results, we established a dose-independent recommendation (3 mg pancuronium bromide and 2.5 mg neostigmine) for animals weighing ≥ 270 kg (TL ≥ ~3.8 m). When applied to 32 adult male crocodiles (BW range: 270–460 kg; TL range: 3.76–4.48 m), the shortest induction time was ~20 min and the longest ~45 min. (4) Conclusions: Pancuronium bromide and its antidote, neostigmine, are effective for the immobilization and reversal of adult male Nile crocodiles (TL ≥ 3.8 m or BW ≥ 270 kg) when given in a weight-independent fashion.

## 1. Introduction

Capturing adult crocodiles is challenging and dangerous due to their significant size, aggressiveness, and potential to cause severe injury to themselves and to operators [1,2,3,4].

Gallamine triethiodide (Flaxedil 20 mg/mL), a nondepolarizing neuromuscular blocker, has been commonly used since the 1970s for immobilizing captive and wild saltwater crocodiles (*Crocodylus porosus*) and Nile crocodiles (*Crocodylus niloticus*) [5,6,7,8,9]. Following the discontinuation of gallamine triethiodide in Australia, pancuronium bromide has empirically proven to be an effective and affordable alternative for immobilizing wild and farmed saltwater crocodiles [4]. Research in mostly juvenile crocodiles (total length [TL] < 2.9 m and body weight [BW] < 246 kg) established a minimum effective dose of 0.019 mg/kg and a safety margin of 0.2 mg/kg for pancuronium bromide, with effective reversal obtained with neostigmine methylsulfate administered at 0.02 mg/kg [10,11]. Limited dose recommendation has, however, been published for adult saltwater crocodiles with a BW ≥ 246 kg despite it being inadvisable to extrapolate a dose from smaller individuals [4,10,11,12]. In addition, dose recommendations for pancuronium bromide have not been published for Nile crocodiles. Here, we report a dosing recommendation using pancuronium bromide and neostigmine methylsulfate for the immobilization of large adult captive Nile crocodiles.

## 2. Materials and Methods

In 2022, a total of 32 adult captive Nile crocodiles were scheduled for transportation from Djerba Explore Park, Djerba, Tunisia to the Dubai Crocodile Park, Dubai, United Arab Emirates. For this purpose, pancuronium bromide (Pavulon, 4 mg/2 mL, Neon Laboratories Ltd., Mumbai, India) was chosen as the drug of choice for its high safety profile, availability, low cost, and ability to be reversed with neostigmine methylsulfate (Stigmine, 0.5 mg/mL, Société Arabe des Industries Pharmaceutiques, Tunis, Tunisia).

Prior to transport, we trialed and fine-tuned a pancuronium bromide dose established for saltwater crocodiles by Bates et al. in subadult and adult Nile crocodiles [10,11]. All of the animals were reversed with neostigmine methylsulfate after one to two hours. On the basis of the results obtained from the trial, we established a dose for animals weighing ≥ 270 kg (TL ≥ ~3.8 m), consisting of 3 mg pancuronium bromide (1.5 mL Pavulon 4 mg/2 mL) and 2.5 mg neostigmine methylsulfate (5 mL Stigmine 0.5 mg/mL). Subsequent to the trial, 32 adult male crocodiles (distinct from those included in the trial group) were immobilized using the new recommendation and shipped to the Dubai Crocodile Park. For crocodiles being transported, the reversal agent was administered by intramuscular injection into the hind limb once immobilization was achieved and the crocodile was secured inside the transport crate (see below).

The depth of neuromuscular blockage was ascertained by prodding the base of the tail with a pole (absence of avoidance reflex), verifying that the mouth remained open in a relaxed position (the presence of the “Flaxedil (i.e., gallamine) reaction” [13]), and inserting a pole inside the mouth (absence of bite reflex). Full recovery was defined as crocodiles moving with a normal gait.

Before attempting injection during the trial and translocation phases, the enclosures were drained of water to eliminate any risk of drowning. Crocodiles were administered pancuronium bromide intramuscularly in the left lateral aspect of the base of the tail using an expandable (90–180 cm) pole syringe (Jabstick, Daninject, Olgas Allé 4, 6000 Kolding, Denmark) equipped with plastic syringes (Jabstick) and 2.0 × 50 mm (14G × 2″) metallic needles (Kruuse A/S, Havretoften 4, DK-5550 Langeskov, Denmark). Only resting crocodiles were targeted to avoid injuring both the crocodile and the operator [14]. To calculate the pancuronium bromide dose, BW was first estimated based on the park’s experience of extrapolating BW from TL. An accurate weight was obtained by weighing crocodiles during immobilization, except for four large crocodiles during the trial, which were too heavy to be weighed on site. All animals were paint-marked immediately after injection to avoid accidental double-dosing. For safety reasons, the jaws were secured with rope. Because nondepolarizing neuromuscular blockers do not abolish sensory input, the head was covered with a towel to reduce sensory stimuli, taking care not to obstruct the nostrils. Body temperature was measured immediately after induction, either using a cloacal thermometer inserted in the proctodeum (Checktemp, Cifec, 12bis rue du commandant Pilot, 92200 Neuilly-sur-Seine, France) during the trial, or a temperature gun (Ketotek, Xiamen Sizhi E-commerce Co., Ltd., 8E Mingyuan Building, 361000 Xiamen, China) aimed at the nuchal region before and during the transport phase [15,16]. Because the duration to induction and the duration of immobilization are expected to be inversely proportional to body temperature [10,11], we only proceeded with the trial and transport phases once ambient temperatures were above 19–20 °C. Following immobilization, crocodiles were handled carefully, kept out of direct sunlight (ambient temperature: 20–24 °C), and regularly doused with water [17].

Crocodiles marked for translocation were transported in individual wooden crates with foam-covered siding. The ambient temperature range measured in the crates was 13–20 °C during road transport and 22–24 °C during air transport. Transport was carried out in two separate shipments (16 animals per shipment) at a one-week interval. All of the transport specifications were in accordance with CITES and IATA (International Air Transport Association) guidelines [18,19]. The two translocations took an average of 43 h, consisting of crating (10 h 00 min and 8 h 15 min), followed by ground (22 h 30 min and 18 h 50 min), air (7 h 00 min and 6 h 40 min), and ground (7 h 10 min and 5 h 20 min) transport. 

## 3. Results

The trial involved one subadult (BW: 54 kg; TL: 2.20 m) and eight adults (BW range: 130–370 kg; TL range: 2.75–4.30 m). The average cloacal temperature was 23 °C (range: 19–26.2 °C). The average induction time was 70 min (range: 20–143 min), and the average recovery time was 22 h and 14 min (median: 105 min; range: 50 min–5 days) (Table 1). The recovery time for three adults having received the highest pancuronium dose (0.011–0.015 mg/kg) was >24 h (two of which required an additional neostigmine injection), whereas the highest dose (0.019 mg/kg) administered to one subadult (54 kg) did not lead to delayed recovery. Four adult crocodiles were subsequently immobilized using a lower dose of 0.006–0.008 mg/kg, leading to a longer average induction time of 109 min (range: 84–143 min) but a shorter average recovery time of 120 min (range: 55–105 min). Based on these initial findings, and considering that the transfers to Dubai would only involve large adult specimens with a BW ≥ 270 kg (TL ≥ 3.8 m), it was decided to substantially reduce the pancuronium bromide dose in adults, deciding on a weight-independent dose of 3 mg pancuronium bromide (1.5 mL Pavulon) and 2.5 mg neostigmine methylsulfate (5 mL Stigmine) independent of BW or TL.

The transport phase involved 32 adult male crocodiles (aged approximately 25 years). The average BW was 365 kg (median: 371 kg; range: 270–460 kg), with an average TL of 4.17 m (range: 3.76–4.48 m). The average nuchal temperature at induction was 25.6 °C (range: 20.8–37.2 °C). The final pancuronium bromide dose range was 0.007–0.011 mg/kg (mean: 0.008 mg/kg, ± SD 0.001). For logistical reasons, individual induction time was not measured; however, the shortest induction time was ~20 min and the longest ~45 min. Recovery time could not be recorded because crocodiles were already inside crates by the time they would have recovered from immobilization. All of the 41 crocodiles included in the trial and transport phase were alive and apparently healthy approximately ten months after the intervention, with no reported instances of anorexia. No injuries during shipment were noted. Approximately ten males displayed mating behavior within a day of arrival at the Dubai Crocodile Park.

## 4. Discussion

Immobilizing agents used in crocodiles include opioids (e.g., etorphine), dissociative anesthetics (e.g., ketamine, tiletamine–zolazepam), alpha-2 agonists (e.g., medetomidine), barbiturates (e.g., pentobarbital sodium), neuroactive steroids (e.g., alfaxalone), neuromuscular blocking agents (e.g., succinylcholine chloride, d-tubocurarine, atracurium besylate, gallamine triethiodide, and pancuronium bromide), and other agents (e.g., propofol, tricaine mesylate) [10,11,13,17,20,21,22,23,24,25,26,27,28,29,30].

Neuromuscular blocking agents (NBAs) can be separated into two classes depending on their mechanism of action. Depolarizing neuromuscular blockers (e.g., succinylcholine chloride) act by binding to postsynaptic cholinergic receptors on motor endplates, causing depolarization and fasciculation, which leads to flaccid paralysis. Nondepolarizing neuromuscular blockers (e.g., d-tubocurarine, atracurium besylate, gallamine triethiodide, and pancuronium bromide) are competitive acetylcholine antagonists that bind to nicotinic receptors on the postsynaptic membrane, blocking acetylcholine binding and thereby preventing motor endplate depolarization. Nondepolarizing neuromuscular blockers can be reversed with cholinesterase inhibitors, such as neostigmine [1,3,5,8,13].

There is a paucity of data on the immobilization of crocodiles using pancuronium bromide. Bates et al. found that the average induction time in juvenile (BL < 2.90 kg) and adult (BL < 90 kg) saltwater crocodiles immobilized with pancuronium bromide was 22 min when given at a dose of 0.02 mg/kg, which was lower than in our cohort [10,11]. The authors also showed that a higher dose of pancuronium bromide (>0.025 mg/kg) did not significantly decrease induction time. In addition, the duration of immobilization (without reversal with neostigmine) was proportional to the pancuronium bromide dose and exercise intensity prior to immobilization (possibly linked to muscle fatigue). In contrast, the duration of immobilization was inversely proportional to body temperature (possibly because of faster drug metabolism) and to induction time. The recovery time was shortened (<5 min) with a higher dose of neostigmine, but the time for the full recovery of all reflexes was not influenced by the dose of pancuronium bromide, sex, body weight, or induction time.

The translocation of 32 Nile crocodiles provided the opportunity to assess our dose recommendation for pancuronium bromide and neostigmine methylsulfate, for which no previous dose recommendations have been published. The results from our trial indicate that it unadvisable to extrapolate dose recommendation by weight, as already hypothesized by Bates et al. and as predicted by the principle of allometric scaling (i.e., body weight is inversely proportional to basal metabolic rate) [10,11]. Although our sample size was small, our observations indicate that extrapolating a dose recommendation established for juveniles in adult crocodiles resulted in a prolonged recovery time despite nearly doubling the recommended neostigmine dose. Using a lower, weight-independent dose of 3 mg pancuronium bromide for crocodiles with TL ≥ 3.8 m or BW ≥ 270 kg (corresponding to 0.007–0.011 mg/kg among transported subjects) resulted in a satisfactory induction time, a result that can be explained by allometric scaling. Based on the limited number of animals included in the trial, we hypothesize that using a higher dose would likely lead to a delayed recovery time. We were, however, unable to assess a lower, weight-independent dose of 2.5 mg neostigmine sulfate (corresponding to 0.006–0.009 mg/kg among transported subjects) because crocodiles were physically restrained and transported in crates for an average of 43 h.

Unlike general anesthetics, NBAs cause flaccid muscle paralysis without sedation or analgesia, thereby reducing the physical and physiologic risks associated with anesthesia, especially during transportation. Monitoring the respiration rate is nevertheless advised because NBAs can affect respiratory function in crocodiles, with a 66% decrease in respiratory rate noted for dose >0.2 mg/kg [4,10]. Our failure to measure the respiratory rate during the field trial constitutes a limitation of our report, although the survival of all 32 crocodiles during long-distance transport by road and air immediately after reversal suggests that respiratory depression, if present, is unlikely to have been severe. 

Animals immobilized with NBA are fully conscious and sensitive to visual, auditory, and tactile stimuli, which can result in stress-induced physiologic reactions such as tachycardia and tachypnea [3,10]. In our case, immobilization was conducted with a minimum of people and reduced noise, and eyes and ears were covered to reduce stress from sensory input [13]. For this reason, any pain-inducing procedures should never be performed when a crocodile is immobilized with NBAs alone.

Physical techniques such as nets, snares, and traps can be used for capturing crocodiles, but they are a frequent source of stress and injury (e.g., fractures, ocular injuries, and skin abrasions), particularly due to fight or flight behaviors that can last until exhaustion (e.g., biting, tail thrashing, and body rolling) [3,4]. Anorexia of 18 months in duration has also been reported in a Mugger crocodile (*C. palustris*) following manual capture [31]. In instances of prolonged struggle (e.g., during protracted capture), physiological stress responses can result in the release of adrenaline and corticosterone, and increased anaerobic glucose catabolism [16,18,19,32]. This can lead to lactic acidemia, capture myopathy, and cardiac dysfunction, sometimes with fatal consequences [13,32,33,34]. Crocodilians, and reptiles in general, lack the ability to rapidly metabolize and correct acidemia, which can be severe [35]; blood pH levels of 6.6–6.8 (reference range: 7.0–7.4) have been recorded in Nile crocodiles after prolonged struggling [3]. Lactate levels, which can be considered as an indicator of anaerobic metabolism and physiological stress in crocodilians, were positively correlated with capture duration (longer handling time or struggling causes an increase in lactate), age (adult crocodilians had significantly higher lactate levels after capture than non-adults), and weight (possibly confounded with handling time because larger animals are generally harder to capture) [34]. Blood biochemistry values in chemically immobilized crocodiles have, however, not been published and might differ from physically caught crocodiles in terms of stress markers (e.g., blood lactate and pH) and physiological parameters (e.g., heart rate and respiratory rate).

## 5. Conclusions

Pancuronium bromide is effective for the immobilization of adult male Nile crocodiles (TL ≥ 3.8 m or BW ≥ 270 kg) when given in a weight-independent fashion. Reversal using neostigmine methylsulfate was equally effective when given in a weight-independent fashion, although the small cohort (*n* = 5) in which reversal was measured precludes any firm conclusions. Systematic studies in a larger population of animals (juveniles, subadults, and adults) are needed to confirm our observations that dose is inversely proportional to body weight and to monitor the effects of pancuronium bromide on physiological parameters and stress markers.

## Figures and Tables

**Table 1 animals-13-01578-t001:** Induction and recovery time for the immobilization of subadult and adult Nile crocodiles (*Crocodylus niloticus*) using pancuronium bromide (Pavulon, 2 mg/mL) and neostigmine (Stigmine, 0.5 mg/mL) based on existing recommendations in juvenile saltwater crocodiles (*Crocodylus porosus*), *n* = 9.

Sex	Body Weight (kg)	Total Length (cm)	Cloacal Temperature (°C)	Pancuronium Bromide (mg/kg)	Pavulon (mL)	Induction Time (min)	Neostigmine (mg/kg)	Stigmine (mL)	Recovery Time
M	54	220	19.0	0.019	0.5	20	0.014	1.50	50 min
F	130	275	20.5	0.015	1.0	30	0.034	8.75 ^§^	5 days
F	172	315	21.6	0.008	0.7	50	0.015	5.00	120 min
M	260	380	22.6	0.015	2.0	45	0.019	10.00 ^§^	2 days
M *	270	390	25.4	0.008	1.1	85	0.009	5.00	105 min
M *	330	410	26.0	0.007	1.1	84	0.008	5.50	90 min
M	353	425	20.4	0.011	2.0	55	0.007	5.00	1 day
M *	370	425	25.8	0.006	1.1	125	0.007	5.00	70 min
M *	370	430	26.2	0.006	1.1	143	0.007	5.00	55 min

* Approximate body weight; ^§^ received two doses.

## Data Availability

Data supporting the reported results can be found at Djerba Explore Crocodile Pars, Djerba, Tunisia: marc.gansuana@gmail.com.

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
