# Peer review of "Pancuronium Bromide for Chemical Immobilization of Adult Nile Crocodiles (Crocodylus niloticus): A Field Study"

_animals, 2023, doi:10.3390/ani13101578_

Round 1

Reviewer 1 Report

Reporting the use of an effective immobilizing agent in a novel species that is both large and potentially dangerous is important. However, there are some features about the study that I felt needed attention or clarification. These are as follows:

1.) It is unclear if any of the adult crocodiles involved in the initial trial were also subsequently transported, or if the two groups (trial and transport) involve individuals that were mutually exclusive. 

2.) Is there a reference for the "Flaxedil reflex"? The average reader may not be familiar with this term. My understanding is that Flaxedil is a brand name for gallamine. 

3.) Although the authors extrapolated a weight-independent dose from the trials, it is worth reporting the actual dose characteristics that were ultimately given to the transport crocodiles (mg/kg). This might provide a more meaningful recommendation to others who wish to immobilize large crocodiles.  

4.) Are cloacal temperatures correlated with nuchal temperatures? Is the temperature data important? Did it influence immobilization or reversal times? 

5.) Although blood pH levels are reported for crocodiles that are physically restrained, they are not reported for immobilized crocodiles. Given the potential physiological complications and stress that could arise from immobilization (including respiratory depression, remaining fully conscious, etc.) is it at all meaningful to include one and not the other as justification for the use of NMA?  

6.) Prolonged post handling anorexia is identified as a complication of physical restraint; however, this study only identifies full recovery from immobilization as moving with a normal gait and long-term survival. Were the crocodiles normal in other respects? 

7.) How were the authors defining "effective" reversal and how can they state it was effective? The recovery times were highly variable for the trial crocodiles, and they were not observed or recorded in the transport crocodiles.   

Author Response

Please see attached pdf containing our answers

Reviewer 2 Report

The reviewer thanks the authors for an overall well written and interesting report. The findings of this report will be a valuable contribution to our field and form a basis for further research regarding save immobilization of crocodiles. Only minor changes are required.

Line 13. Change carry to carries

Line 30. Add bromide after pancuronium

Line 88. Use active substance instead of brand name for ease of extrapolation

Line 125. Change ‘we decided’ to ‘it was decided’

Line 127. 0.25 or 2.5 mg / 0.5 mL or 5 mL Stigmine? Change accordingly for each dose mentioned (Cfr. Lines 36; 75-76; 213).

Lines 175-198. Move paragraph lines 183-192 ‘NBAs can be separated…  such as neostigmine’ up to follow after line 175 ‘…rapid induction time and ease of reversal.'

Lines 203-204. Could it be interesting to add dose per body surface or a similar allometric scaling value in Table 1 to compare induction and recovery times to these values?

Lines 207-208. Use active substance and mg dose for ease of extrapolation if other formulations are marketed in the future. Change 1.5 mL Pavulon to 3 mg pancuronium bromide.

Line 213. Use active substance and mg dose for ease of extrapolation if other formulations are marketed in the future. Change 5 mL Stigmine to 2.5 mg neostigmine.

As mentioned in lines 193-198, while saver from physical damage, stress remains a concern. Perhaps the authors could formulate an advice/suggestion for future research to include monitoring of physiological parameters and other stress markers or suggestions of possible drug combinations to further reduce stress either here (193-198) or in their final statement?

Author Response

(The authors gave the same response as above.)

Reviewer 3 Report

Paper well written and pointing out an important topic.

A few modifications are advisable:

-          Line 95 except instead of expect (typo).

-          Line 101 add details of the cloacal temperature measurement (how deep the probe went in the cloaca? Did it overcome the proctodeum?)

-          Line 103 add citation to the temperature check method

-          Line 106-112 add the transport temperature of the animal (environmental).

-          Material and methods: it is unclear when the neostigmine is given to the animals, better state the timing between induction and reversal

-          I would add a table to include descriptive statistics: mean and sd length, mean and sd estimated weight, mean and sd dosage and mean recovery time (if possible, if not clearly state it).

Discussion shall be reduced and better presented following the standard order: relevant scientific knowledge on the topic, results comparison, conclusions (in this case I would remark the dosage, and timing of the pancuronium and neostigmine)

-          Bibliography needs to be better uniformed: doi is missing or not uniform in many citations (http present or absent) overall a lot of grey literature cited, which is understandable due to the scarcity of papers on the topic. 

-  

Author Response

(The authors gave the same response as above.)

Round 2

Reviewer 1 Report

This manuscript has been significantly improved and the content is clear and concise.